# Metabolic Response in Endothelial Cells to Catecholamine Stimulation Associated with Increased Vascular Permeability

**DOI:** 10.3390/ijms23063162

**Published:** 2022-03-15

**Authors:** Adrián López García de Lomana, Arnar Ingi Vilhjálmsson, Sarah McGarrity, Rósa Sigurðardóttir, Ósk Anuforo, Alexía Rós Viktorsdóttir, Aris Kotronoulas, Andreas Bergmann, Leifur Franzson, Haraldur Halldórsson, Hanne H. Henriksen, Charles E. Wade, Pär Ingemar Johansson, Óttar Rolfsson

**Affiliations:** 1Center for Systems Biology, University of Iceland, 101 Reykjavík, Iceland; adrian@hi.is (A.L.G.d.L.); aiv3@hi.is (A.I.V.); sarahm@hi.is (S.M.); rosasig1993@gmail.com (R.S.); adaanuforo@hotmail.com (Ó.A.); alexiarosv@gmail.com (A.R.V.); akotronoulas@gmail.com (A.K.); andreasbergmannhome@gmail.com (A.B.); 2Landspítali-Háskólasjúkrahús, 101 Reykjavík, Iceland; leifurfr@landspitali.is; 3Medical Department, University of Iceland, 102 Reykjavík, Iceland; haralhal@hi.is; 4Center for Endotheliomics CAG, Rigshospitalet, 2200 Copenhagen, Denmark; hanne.hee.henriksen@regionh.dk (H.H.H.); per.johansson@regionh.dk (P.I.J.); 5Center for Translational Injury Research, The University of Texas Health Science Center, Houston, TX 77030, USA; charles.e.wade@uth.tmc.edu

**Keywords:** metabolomics, catecholamines, vascular permeability, endotheliopathy, major trauma

## Abstract

Disruption to endothelial cell homeostasis results in an extensive variety of human pathologies that are particularly relevant to major trauma. Circulating catecholamines, such as adrenaline and noradrenaline, activate endothelial adrenergic receptors triggering a potent response in endothelial function. The regulation of the endothelial cell metabolism is distinct and profoundly important to endothelium homeostasis. However, a precise catalogue of the metabolic alterations caused by sustained high catecholamine levels that results in endothelial dysfunction is still underexplored. Here, we uncover a set of up to 46 metabolites that exhibit a dose–response relationship to adrenaline-noradrenaline equimolar treatment. The identified metabolites align with the glutathione-ascorbate cycle and the nitric oxide biosynthesis pathway. Certain key metabolites, such as arginine and reduced glutathione, displayed a differential response to treatment in early (4 h) compared to late (24 h) stages of sustained stimulation, indicative of homeostatic metabolic feedback loops. Furthermore, we quantified an increase in the glucose consumption and aerobic respiration in endothelial cells upon catecholamine stimulation. Our results indicate that oxidative stress and nitric oxide metabolic pathways are downstream consequences of endothelial cell stimulation with sustained high levels of catecholamines. A precise understanding of the metabolic response in endothelial cells to pathological levels of catecholamines will facilitate the identification of more efficient clinical interventions in trauma patients.

## 1. Introduction

The endothelium is a whole-organism level organ with an estimated weight of 1 kg and covering a surface area of 4000 to 7000 m^2^ in adult humans [1]. Endothelial cells (ECs) form a monolayer of squamous cells lining the interior of blood and lymphatic vessels [2]. The endothelium regulates vital processes, including oxygen, nutrient and hormone supply to tissues [3], coagulation and haemostasis [4], vascular tone and blood vessel permeability [5], immune cell trafficking and inflammation [6,7] and tumour metastasis [8]. Further, the phenotypic diversity of ECs is paramount and strongly influenced by vascular beds in healthy conditions [9,10].

Metabolic function and regulation in ECs are particularly important for endothelial function [11]. For example, glycolysis, which is the primary energy-producing mechanism in ECs [12], promotes migration and proliferation during angiogenesis [13,14]. The glutamine metabolism is also strongly linked to EC proliferation [15]. Fatty acid oxidation (FAO) is a metabolic pathway uniquely relevant for ECs. ECs rely on FAO to produce nucleotides sustaining the TCA cycle. 

Moreover, FAO produces NADPH used to generate reduced glutathione to control oxidative stress and provide reducing power to enzymes key for vascular homeostasis—nitric oxide synthase 3 (eNOS) and prostaglandin G/H synthase 1 (PTGS1) [16]. Indeed, there is a direct link from FAO to vascular permeability as inhibition of the rate-limiting enzyme of FAO results in hyperpermeability in vitro and blood vessel leakage in vivo [17]. Nitric oxide (NO), a broad and potent regulator of vascular homeostasis, is a product of the arginine metabolism [18]. 

A relevant proposal is the idea that the EC phenotype can be directly controlled through metabolic perturbations, bypassing upstream signalling, which has been recently demonstrated experimentally by promoting angiogenesis through *PFKFB3* overexpression, even overriding counteracting anti-angiogenic Notch signals [14]. Altogether, the mechanistic understanding of how metabolism influences EC phenotypes represents a promising therapeutic opportunity to control human health to disease transitions.

Endothelial dysfunction is a systemic pathological condition characterised by increased vascular permeability and increased oxidative stress that results in vascular tone loss. Endothelial dysfunction is of clinical significance to a large broad plethora of diseases, including cancer [19], diabetes [20], atherosclerosis [21], COVID-19 [22] and acute critical illness (ACI) [23]. 

In particular, ACI is a disparate group of conditions, including trauma, sepsis, myocardial infarction and post-cardiac arrest syndrome, with a roughly estimated incidence of 60 million patients worldwide each year [24], who are united by endothelial dysfunction associated with elevated sympathoadrenal signalling. Sustained catecholamine upregulation provokes systemic adverse effects characterised by endothelial permeability and glycocalyx loss, in both in situ and in vitro models [25,26]. Classical clinical biomarkers of endothelial dysfunction in ACI patients are circulating levels of the adhesion molecules VCAM-1, ICAM-1 and E-selectin and blood coagulation factors, such as the von Willebrand factor and soluble thrombomodulin. 

Other novel molecular biomarkers currently emerging are endoglin, endocan and syndecan, among others [27,28]. ACI has been shown to have effects on both the protein and glucose metabolism [29,30], leading to the inclusion of glucose and lactate measurements in control treatment guidelines for ACI [31]. Indeed, blockers of beta-adrenergic signalling have been shown to mitigate critical illness in multiple clinical trials [32], yet a deep mechanistic understanding of the molecular regulation leading to the onset of this condition is lacking, particularly for the role of endothelial metabolism, with the potential to open new avenues for novel drug interventions.

In this study, we report a metabolomics-centred approach capturing metabolic phenotypes of catecholamine-induced endothelial dysfunction in human umbilical vein endothelial cells (HUVECs). We observed increased vascular permeability and glycocalyx loss in HUVECs stimulated with an equimolar ratio of adrenaline and noradrenaline recapitulating an in vitro model of ACI. Following validation of the cell culture model, we determined the nutrient uptake and secretion rates and observed clear differences in the glucose and lactate exchange rates upon treatment stimulation. 

Next, we measured the effects of catecholamines in glycolysis and aerobic respiration in HUVECs. Finally, we performed untargeted metabolomic analysis alongside ^13^C glucose and ^15^N glutamine pulse-chase experiments to define changes in metabolism following adrenal stimulation. Together, these results constitute a quantitative catalogue of metabolic outcomes in HUVECs upon treatment with sustained high levels of catecholamines, providing queues for a better understanding of the metabolic shifts that occur in endothelial dysfunction.

## 2. Results

### 2.1. Catecholamine Stimulation Leads to Increased Vascular Permeability and Glycocalyx Loss in HUVECs

Towards a better understanding of the impact of catecholamines on vascular permeability, we exposed HUVECs to a wide range of an equimolar ratio of adrenaline and noradrenaline (0, 0.5, 5 and 50 μM). We assessed cell culture barrier function by measuring the transendothelial electrical resistance (TEER; see Methods). We sampled cell cultures at 0, 1, 4 and 24 h (time points T0, T1, T4 and T24, respectively). The experimental design is succinctly illustrated in Figure 1A. We found that catecholamines increased the vascular permeability in vitro. We observed a permeability increase upon treatment with a 5 µM equimolar catecholamine mix at both 4 and 24 h after treatment (Mann–Whitney *U* test; *p* < 0.03 and *p* < 0.05, respectively; orange markers in Figure 1B). 

For this concentration, we also found a substantial permeability increase at 1 h after treatment; however, due to the lack of response in the HUVECs of one donor, the differences for this group remained statistically not significant with respect to the control samples. Lower catecholamine concentrations, i.e., 0.5 µM, did not induce significant permeability changes at any time (green markers in Figure 1B). As a reference, tumour necrosis factor-alpha (TNF-α) induced a stronger permeability increase compared with the 5 µM catecholamine treatment but at a considerably later time—24 h after treatment (Mann–Whitney *U* test; *p* < 0.03; sky blue markers in Figure 1B). Upon these results, we concluded that stimulation with high concentrations of catecholamines induced increased vascular permeability.

The endothelial glycocalyx (eGC) regulates a myriad of physiological roles, including vascular permeability [33]. Thus, we sought to investigate if the observed permeability increase in HUVECs upon catecholamine stimulation is accompanied by eGC loss or shedding. We generated transmission electron microscopy (TEM) images of HUVECs after four hours of exposure to different catecholamine levels confirming a fairly representative eGC morphology [34]. We observed a dose–response relationship between catecholamine concentration and eGC loss (Figure 1C). This observation aligns well with clinical findings in ACI patients associating circulating catecholamines with eGC damage [35,36]. Conclusively, we interpret that catecholamine stimulation induces both the increase of permeability and loss of eGC in HUVECs in vitro, modelling previous clinical observations.

### 2.2. Catecholamine Stimulation Increases Glucose Uptake and Lactate Secretion in HUVECs

We sought to investigate whether catecholamine stimulation in HUVECs results in a concomitant alteration in metabolic function together with observed increased vascular permeability and glycocalyx loss. We measured the cellular transport rates (uptake and secretion) of selected metabolites present in the media and quantified the extracellular flux rates of glucose, lactate and pyruvate and a broad panel of amino acids using different methods—a blood gas analyser and enzymatic kit assays (see Section 4 for details). 

We observed that treatment resulted in an increase in the glucose uptake rate, an increase in lactate secretion and no apparent trend in the pyruvate extracellular flux rates (Figure 2). Furthermore, we found no consistent pattern in the amino acid transport rates associated with treatment (Appendix A). Overall, these results indicate that catecholamine stimulation provokes an increase in glucose utilisation in ECs, as supported by increased glucose uptake and increased lactate secretion.

### 2.3. Catecholamine Stimulation Increases Glycolysis and Aerobic Respiration in HUVECs

Provided the above-presented evidence on glucose consumption and lactate production changes upon catecholamine treatment, we sought to investigate whether treatment with catecholamines in HUVECs results in further alterations in metabolic function. In particular, we assessed the effect of catecholamine stimulation on glycolysis, ATP production and respiratory function. Towards this goal, we performed a Seahorse assay with cultured HUVECs exposed to different levels of catecholamines, both at different concentrations and exposure times (see Methods for details). The seahorse assay quantifies both extracellular acidification rate (ECAR), which reflects the glycolytic cell function and oxygen consumption rate (OCR), which is an indicator of mitochondrial respiration. 

Furthermore, the sequential use of three chemical inhibitors of the electron transport chain components facilitates the understanding of metabolic changes derived from mitochondrial respiration—first, oligomycin inhibits ATP synthase resulting in a drop in OCR that can be associated with respiratory ATP production; then, carbonyl cyanide-p-trifluoromethoxyphenylhydrazone (FCCP) disrupts the mitochondrial membrane potential allowing unrestricted electron flow that causes a large increase in OCR as a proxy for maximal respiratory capacity; and finally, the addition of both rotenone and antimycin A, inhibitors of complex I and III, respectively, blocks mitochondrial respiration completely, revealing non-mitochondrial respiration by quantifying the remaining OCR.

Using this approach, we found that catecholamine stimulation increases glycolysis. Following both a 4 and 24-h pretreatment, we observed an increase in the basal glycolytic function in a dose–response relationship (Mann–Whitney *U* test; at least *p* < 0.008 for any comparison against control; Figure 3A). Basal ECAR increased from 3.36 mpH × min^−1^ × cell^−1^ in control cultures to 6.54 and 9.36 mpH × min^−1^ × cell^−1^ in 4-h pre-treated cultures with 0.5 μM and 5 μM catecholamines, which represents a 1.95- and a 2.79-fold increase in glycolytic activity, respectively. We observed a similar trend for cultures pre-treated for 24 h. 

Next, we identified a clear pattern of elevated mitochondrial respiration function upon catecholamine stimulation (Figure 3B–E). Respiration-based ATP production and both basal and maximal respiration increased following a dose–response relationship. We observed a qualitatively consistent response pattern in both HUVECs pre-treated for 4 and 24 h, just smaller in magnitude in the former (Figure 3B,C). Focusing in the 24 h pre-treated cells, basal respiration increased from 91 to 145 and 200 pmol × min^−1^ × cell^−1^ upon 0.5 and 5 μM catecholamine pretreatment, which represents a 1.58- and 2.18-fold-change increase, respectively (Mann–Whitney *U* test; at least *p* < 0.003 for any comparison against control; Figure 3D). 

Similarly, maximal respiration also increased in a dose–response relationship from 202 to 332 and 468 pmol × min^−1^ × cell^−1^ upon 0.5 and 5 μM catecholamine pretreatment, which represents a 1.64- and 2.31-fold-change increase, respectively (Mann–Whitney *U* test; *p* < 0.003 for any comparison against control; Figure 3E). Identified as a drop in OCR after oligomycin injection, respiration-based ATP production also substantially increased from 42 to 68 and 98 pmol × min^−1^ × cell^−1^ upon 0.5 and 5 μM catecholamine pretreatment, respectively (OCR drop after oligomycin in Figure 3C). However, the ratio of respiration-based ATP production over basal OCR did not change upon treatment, staying constant at around 46–49%. Taken together, these results point to increased metabolic activity in glycolysis and mitochondrial respiration in HUVECs upon catecholamine stimulation. 

### 2.4. Catecholamine Dose-Responding Metabolites in HUVECs

Towards a better understanding of the impact of catecholamines on endothelial metabolism, we followed the same experimental design as described in Figure 1A. Briefly, we exposed HUVECs to a wide range of equimolar ratios of adrenaline and noradrenaline at different concentrations (0, 0.5, 5 and 50 μM). We sampled cell cultures at 0, 4 and 24 h (time points T0, T4 and T24, respectively) and quantified their metabolomes using untargeted metabolomics (see Section 4). We detected 1230 metabolic features (*m*/*z* peaks) that we quantified across all samples. This dataset represents an ideal resource to mine for metabolic changes in ECs upon catecholamine stimulation in vitro. 

As we are interested in the biological response to catecholamines, we investigated which metabolic features responded consistently to treatment in a dose–response manner. We selected those features that had a substantial difference upon stimulation (abs log_2_ FC > 1; FC, fold change) and exhibited a strong significant correlation with catecholamine levels (Pearson correlation coefficient > 0.8 and *p* < 0.05). We found a set of 46 features that complied with these rules in either T4 or T24. We considered that this set of 46 metabolic features fairly maps to the biologically relevant metabolic changes in our experimental design.

We hypothesised that the metabolic state of early (4 h) versus late (24 h) catecholamine-stimulated ECs may be different. Therefore, we sought to reveal consistent groups of dose–response metabolites with differential patterns at early and late time points. Taking the 46 metabolic features identified earlier, we used agglomerative clustering and the Davies–Bouldin index to quantitatively assess the optimal clustering partition (Appendix A). 

We found our dose-responding metabolic features best grouped together into seven clusters, each with a distinct response pattern (Figure 4A). The largest cluster consisted of features that decreased at T4 but bounced back at T24. Another relatively large cluster of metabolic features increased at both T4 and T24. Interestingly, we found a cluster that increased at T4 but decreased at T24 and another cluster with the inverse pattern. Additionally, we found other smaller clusters that either increased at early times or decreased at late times only. This analysis points to a rich metabolic response in ECs to both dose and exposure time upon catecholamine stimulation.

After manual annotation of the set of 46 metabolic features, we identified representative metabolites within the particular temporal patterns. As expected, we observed an evident increase of intracellular adrenaline derivatives. Both adrenaline sulphate and metadrenaline sulphate dynamics perfectly matched the experimental design reassuring successful cell stimulation (Figure 4B). 

Catecholamine sulfonation represents a physiological inactivation mechanism, as free catecholamines are short-lived in plasma (1–3 min) [37]. While this process has been extensively studied, catecholamine sulphoconjugates source tissue is controversial, with current indications for the liver, platelets and the gastrointestinal tract [37]. Our finding of a clear dose–response in conjugated adrenaline upon stimulation indicates that ECs are metabolically capable of adrenaline sulfonation, suggesting the endothelium as a physiologically relevant source of the circulating sulphoconjugated catecholamines in the bloodstream.

Once we confirmed cellular catecholamine incorporation, we investigated which other metabolite changes derived from treatment. As early as four hours after stimulation, we identified the depletion of reduced glutathione (GSH) and an increase in hypothiocyanite, metabolites consistent with changes in redox homeostasis (Figure 4B,C). GSH is a major protectant of reactive oxygen species (ROS) in ECs [38], whose depletion has been associated with oxidative stress and concomitant vascular dysfunction [39]. Hypothiocyanite is a mild oxidant with particular effects on endothelial function as it blocks apoptosis in HUVECs via caspase-3 inhibition [40]. 

This apparent increase in oxidative stress continues at later times with an increase in ascorbate and a decrease in oxoproline. Ascorbate accumulated up to eightfold 24 h after catecholamine stimulation in a dose–response manner (Figure 4D). Increased ascorbate may explain why GSH levels bounced back to normal levels at 24 h. Such a biphasic pattern for GSH recovery has been previously described in HUVECs under H_2_O_2_ exposure [41]. Additionally, the oxoproline depletion (Figure 4E) can also be interpreted along the same lines, as it is a precursor of glutathione biosynthesis. Indeed, oxoproline plasma levels have been proposed as a biomarker for oxidative stress and poor prognosis in cardiovascular disease [42]. Altogether, these changes point to an increase in oxidative stress in HUVECs upon exposure to catecholamines. 

Further, we investigated if additional changing metabolites aligned with other key endothelial functions. We identified an early decrease of N-acetylglutamate and a late arginine increase upon treatment, indicative of alterations in the NO production pathway (Figure 4C,D). N-acetylglutamate is an allosteric regulator of carbamoyl phosphate synthetase I, the rate-limiting enzyme in arginine biosynthesis; more evidently, arginine acts as a substrate for NO production in ECs [43]. Independently, it has been previously reported that adrenaline increases endothelial nitric-oxide synthase activity [44,45] and sustained increase of L-arginine uptake via NO-mediated membrane hyperpolarization [46]. These results corroborate previous studies that demonstrated increased NO production in HUVECs upon catecholamine treatment [45,47]. 

We found other metabolites that consistently responded to catecholamine stimulation, yet their physiological implications are more challenging to circumscribe. At four hours after treatment (Figure 4A,B), we observed an increase of inorganic phosphate and a decrease of phosphatidylinositol, which can be possibly linked to Akt signalling activation. Further, we identified an increase in cholesterol derivatives and a decrease of lysophosphatidylethanolamine 24 h after stimulation, which suggests changes in lipid metabolism or membrane remodelling (Figure 4D,E). Overall, we uncovered a rich pattern of dose-responding metabolites that point to metabolic remodelling of ECs upon catecholamine stimulation.

### 2.5. Catecholamine Stimulation Does Not Reroute Core Metabolism in HUVECs

ECs reroute their metabolism to sustain homeostasis across various signalling stimuli [16]. Given the observed metabolite changes in the glutathione-ascorbate cycle and the nitric oxide biosynthesis pathway, we sought to investigate if HUVECs reprogram their metabolism upon catecholamine treatment. Thus, taking a similar experimental design as described earlier (Figure 1A), we performed ^13^C and ^15^N labelling experiments using glucose-1,2-^13^C_2_ and glutamine-^15^N_2_ (see Section 4 for experimental details). 

First, we investigated if catecholamine treatment affected the carbon flux direction downstream of glycolysis. We observed no significant differences (Mann–Whitney *U* test; *p* > 0.05) in the mean isotopic enrichment of extracellular lactate from glucose-1,2-^13^C_2_ at either 4 h or 24 h (Figure 5A). A 50:50 mixture of unlabelled glucose:glucose-1,2-^13^C_2_ can result in the theoretical maximum ^13^C mean enrichment of 16.5% in extracellular lactate. After 4 h, we observed a mean enrichment in extracellular lactate of 7.0%, while, after 24 h, it increased to 13.5%, irrespective of treatment. Assuming steady state at 24 h, and accounting for the lactate already present in the media, these data suggest that 86.7% of the carbons in lactate originated from glucose, which is consistent with previous reports [15]. 

Importantly, while catecholamines increase glucose consumption (as we presented in Figure 2), ^13^C-labelled extracellular lactate measurements indicate no rerouting of glucose carbons due to treatment. Next, we compared individual isotopologue enrichments in extracellular lactate in order to calculate the percentage of carbons channelled into the pentose phosphate pathway (PPP) [48]. We found that ~2% of glucose carbon is directed into the PPP, and this ratio remained constant irrespective of catecholamine treatment (Mann–Whitney *U* test; *p* > 0.05; Figure 5B). Together with glycolysis and the PPP, the tricarboxylic acid (TCA) cycle is another major central carbon metabolism pathway in the cell. Thus, we also searched for potential metabolic rerouting to or from the TCA cycle. 

We quantified ^13^C enrichments in citrate, malate and glutamate. We found no significant changes (Mann–Whitney *U* test; *p* > 0.05) in carbon enrichment upon catecholamine stimulation (Figure 5C). Untreated cultures at 24 h after incubation showed a 4.48% ^13^C mean enrichment in glutamate, which, considering the labelled media used, represents a carbon contribution of 17.92% from glucose to glutamate, in accordance with previous experimental evidence in ECs [15]. Conclusively, our results show no rerouting of glucose-derived carbons across the major carbon metabolism pathways, despite the increased glycolytic rate of HUVECs upon catecholamine hit.

The nitrogen metabolism plays a central role in endothelial cell biology. Thus, we investigated whether catecholamines may result in nitrogen metabolism remodelling in HUVECs. We fed HUVECs with a 50:50 mixture of unlabelled glutamine:glutamine-^15^N_2_ in the growth media. We quantified the mean ^15^N enrichments in glutamine, aspartate and glutamate (Figure 5D). 

We found no significant differences (Mann–Whitney *U* test; *p* > 0.05) in ^15^N mean enrichment upon catecholamine treatment, indicative of no pathway rewiring with respect to nitrogen metabolism. These results align well with our prior observation on no significant changes in glutamine uptake due to treatment (Appendix A). We further quantified both ^13^C and ^15^N enrichments on GSH and we found again no significant changes (Mann–Whitney *U* test; *p* > 0.05) upon catecholamine stimulation (Figure 5E,F). Altogether, these results suggest that major nitrogen pathways downstream of glutamine are not rerouted following catecholamine treatment. 

Finally, due to the general relevance of the nucleotide synthesis pathway, and particularly, the role of nicotinamide adenine dinucleotide (NAD) in endothelial dysfunction [49], we quantified the carbon flux directed into this cofactor and a selected set of nucleotides. We found no significant changes (Mann–Whitney *U* test; *p* > 0.05) in carbon incorporation in NAD (Figure 5G); or in the nucleotides, adenosine monophosphate, cytidine monophosphate and thymine monophosphate (data not shown). 

Yet, we found a negative linear relationship (Pearson correlation coefficient *r* = −0.801; *p* < 0.01) between treatment and carbon incorporation into inosine; nevertheless, this effect is small—less than 2% ^13^C mean enrichment difference between control and the highest level of catecholamines tested (Figure 5H). Collectively, these results indicate no consistent diversion of carbon into nucleotide synthesis and salvage pathways following stimulation. 

## 3. Discussion

The main goal of this study was to identify the metabolic response of ECs in an in vitro model of ACI. In summary, we quantified the metabolic changes occurring in HUVECs upon sustained high levels of catecholamines that resulted in increased vascular permeability and glycocalyx loss. Cells became more glycolytic and increased respiration. Untargeted metabolomics revealed a rich metabolic response to treatment dose and time, revolving around two major axes: oxidative stress and NO metabolism. Using isotopic tracers, we showed that no apparent flux rerouting of glycolysis and the TCA cycle takes place following the catecholamine hit. 

eGC loss and vascular permeability are two factors associated with patient mortality, and these two conditions have been reported in clinical environments triggered by high levels of catecholamines [50,51]. Thus, we are confident that our experimental model is representative of ACI as we observed these two hallmarks in our system. 

Upon treatment, HUVECs increased respiration and became more glycolytic, which reflects a higher cellular energy demand, as these two major metabolic pathways govern ATP production. A partially similar pattern was recently observed in HUVECs upon inflammatory activation [52]. IL-1β-stimulated cells increased their oxygen and glucose consumption upon treatment. IL-1β-stimulated cells did not show a concomitant increase in lactate accumulation following increased glycolysis, contrary to our observations in catecholamine-stimulated cells. 

In the case of IL-1β-stimulated cells, it is apparent that there is a metabolic rearrangement that further steers pyruvate into the TCA cycle instead of converting it into lactate. Furthermore, Ziogas et al. [53] recently described a parallel mechanistic scenario, such that pharmacologic blockade of glycolysis prevented histamine-induced vascular hyperpermeability. In the case of catecholamine stimulation, our observations point to a general upregulation of both glycolysis and respiration without redirecting metabolic fluxes. 

There is a substantial body of work associating oxidative stress with endothelial dysfunction [54]. Particularly in the context of ACI, endothelial cells respond to environmental stimuli with a fundamental impact on metabolism. Our findings on metabolic alterations in oxidative stress match previously observations in ACI patients [55]. Moreover, we quantified changes in N-acetylglutamate and arginine, both key metabolites of the NO production pathway, which is an additional pillar of vascular homeostasis. Remarkably, we found a distinct response at 4 h compared to 24 h after catecholamine presence, indicative of metabolic feedback loops driving the cell system into a different state. Altogether we established a catalogue of metabolic perturbations linking high sustained levels of adrenaline to endothelial dysfunction. 

A remarkable finding of our work is the lack of apparent rerouting of glucose-derived carbons across the major carbon metabolism pathways, despite the increased glycolytic rate upon catecholamine hit. We propose as future work to build genome-scale condition-specific predictive models of metabolic flux [56]. The experimental validation of predicted putatively differential fluxes would represent a more successful approach to capture eventual catecholamine-induced flux re-arrangements, particularly in the lipid metabolism.

A working hypothesis motivating our work is that the diverse variety of causes behind ACI, all converge into sympathoadrenal hyperactivation. Sustained high levels of catecholamines drive the endothelium into a dysfunctional state. Indeed, endotheliopathy biomarkers are predictors of mortality in trauma patients [57]. How trauma-induced endothelial dysfunction develops into organ failure and ultimately death is not completely understood [23,58], particularly the metabolic transitions occurring in ECs. 

Therefore, quantifying the metabolic signatures of this process is a valuable roadmap towards finding rational interventions for better treatment of ACI patients. Indeed, novel strategies targeting metabolic regulatory genes have been proved successful against endotheliopathy, e.g., Pfkfb3 knockout protected from pulmonary oedema upon lipopolysaccharide-shock via reduced glycolysis and decreased concomitant endothelial permeability, vascular cell adhesion and immune infiltration [59]. We envision this strategy as a potentially valuable target against catecholamine-induced endothelial dysfunction.

A limitation in our study is the use of both adrenaline and noradrenaline in combination, meaning that we were unable to disentangle their specific effects. While this is a limitation towards a mechanistic understanding, we chose this approach as it is more realistic for the in vivo response. In our study, we did not assess the potential changes in the proteome and transcriptome, which also represents a limitation of our work. 

For example, a deeper mechanistic view would arise from the determination of the gene and protein expression changes responding to the experimental perturbation that is further associated with the observed metabolic changes. It would be particularly interesting to determine whether a small set of transcription factors are able to drive the cell into a new transcriptional state, concomitantly arriving in a different metabolic state [60]. Another limitation of this study is the large variability across donors on proliferation changes due to treatment. 

While we quantified cell numbers in cultures, a higher temporal resolution and more biological replicates would have shed light on more conclusive results on the effects of catecholamine treatment on HUVECs proliferation. Finally, our experiments did not investigate the potentially different responses of HUVECs under different flow conditions. Indeed, HUVECs respond to local flow conditions, modulating their transcriptome [61] and interactions with other vascular structures through cell adhesion [62] and NO production [63].

Unfortunately, the media we used for labelling experiments, EGM-2, already contained unlabelled glucose in it. Therefore, label incorporation from glucose-1,2-^13^C_2_ did not yield high isotopologue enrichments in downstream metabolites. Kim et al. [63,64] circumvented this problem by using DMEM media instead, which is a defined media, allowing the eventual total depletion of glucose and glutamine if desired. Nevertheless, by accounting for tracer purity, we found that the percentage of lactate and glutamate carbons derived from glucose were consistent with those previously reported. 

Future directions of this work revolve around three major axes: (1) the natural extension of this approach to other types of ECs, given the profound differences between microvascular and macrovascular ECs, the large phenotypic diversity of ECs in general and, ultimately, into in vivo experimental models; (2) the determination of metabolic biomarkers of endothelial dysfunction of clinical relevance; and (3) ultimately the exploration of endothelium-specific drugs beyond adrenergic receptor antagonists with potential to alleviate and eventually protect from the detrimental physiologic consequences of sustained high levels of catecholamines. 

Conclusively, this study represents a reference map for metabolic remodelling in ECs upon sustained high levels of catecholamine stimulation, providing new insights to better understand the mechanisms of endothelial dysfunction and paving the way towards novel, rational and more efficient therapeutic interventions against ACI. 

## 4. Methods

### 4.1. Cell Culture

HUVECs were harvested using the method of Jaffe et al. [64,65] from samples obtained at the Landspitali University Hospital in Reykjavik, Iceland. These experiments were performed in accordance with the approval granted by the National Bioethics Committee of Iceland and Icelandic Data Protection Authority, following their guidelines on informed consent. 

Cells were collected by collagenase digestion and seeded in 25 cm^2^ flasks with 5 mL of Endothelial Cell Growth Medium (EGM-2) (Lonza, Basel, Switzerland), containing 2% foetal bovine serum and antibiotics (penicillin, 100 units/mL and streptomycin, 100 μg/mL Thermo-Fisher Scientific, Waltham, MA, USA). The medium was changed 24 h after seeding and then every 48 h until the cells were observed to be confluent (~5 days); they were then used or sub-cultured to no more than Passage 3. 

During the experiments, cells were not subcultured for further use, so each experiment uses HUVECs from different donors. For each experiment, cells were seeded onto 35 mm^2^ culture plates in 2 mL EGM-2. At confluence, a media change into EGM-2 media containing 0.5, 5 or 50 μM of adrenaline and noradrenaline (Sigma-Aldrich) at a 1:1 ratio was added (T = 0). Cells were then incubated for 4 or 24 h. After incubation, cells and media were collected and stored at −80 °C for downstream biochemical analysis further described below. 

For isotope labelling experiments, HUVECs were cultured to confluence as above and then had their medium changed to EGM-2 containing 5 mM glucose-1,2-^13^C_2_ or 10 mM glutamine-^15^N_2_ (Cambridge Isotopes, Tewksbury, MA, USA) with and without catecholamines at the aforementioned concentrations. EGM-2 media was not available in glucose or glutamine depleted form at the time; therefore, we added the labelled metabolites to the media resulting in a 50:50 mixture of labelled-unlabelled metabolites (either glucose or glutamine). The total concentration in the media for glucose or glutamine was double compared to the other conditions. For each experiment, we used one well to determine the cell number by trypan blue exclusion and cell counting in a haemocytometer.

### 4.2. Endothelial Cell Culture Permeability Assay

We assessed cell culture barrier function by measuring the transendothelial electrical resistance (TEER) using the Epithelial Volt/Ohm Meter 3 (World Precision Instruments, Sarasota, FL, USA) instrument as described in the manual. Briefly, we grew HUVECs to confluence on 0.1% gelatine-coated transwell cell culture membrane inserts of 4 µm pore size and 6.5 mm diameter (Corning) in endothelial growth medium (EGM-2; Lonza, Basel, Switzerland) and serum-starved them for two hours in endothelial basal medium (EBM-2; Lonza, Basel, Switzerland) before treatment. A change in TEER across the cell monolayer indicated increased or decreased the paracellular permeability.

### 4.3. Seahorse Assay

We used a Seahorse XFe-96 metabolic extracellular flux analyser (Seahorse Biosciences, North Billerica, MA, USA) to measure the oxygen consumption rate (OCR) and the extracellular acidification rate (ECAR). At 48 h prior to Seahorse assay, HUVECs were seeded into a 96-well XF cell culture microplate at a density of 60,000 cells/well. At 24 h prior to the assay, the Agilent Seahorse XFe-96 sensor cartridge was hydrated in a CO_2_-free incubator in an Agilent Seahorse XF Calibrant (Seahorse Biosciences, North Billerica, MA, USA). 

At 24 h and 4 h prior to the assay, catecholamines at either 0.5 or 5 μM (1:1, adrenaline:noradrenaline mix) were added to culture media. At 2 h prior to the assay, the cells were counted with PrestoBlue Cell viability reagent (Thermo, Waltham, MA, USA), and the growth medium was replaced with Agilent Seahorse XF DMEM Medium pH 7.4 (Seahorse Biosciences, North Billerica, MA, USA) with added catecholamines at the respective concentrations. 

The cell culture plate was then placed in a CO_2_-free incubator for 1 h. Inhibitors were then added to the Agilent Seahorse XFe-96 sensor cartridge; oligomycin (1.5 μM; Sigma-Aldrich, St. Louis, MO, USA), carbonyl cyanide-p-trifluoromethoxyphenylhydrazone (FCCP) (1.5 μM; Sigma-Aldrich, St. Louis, MO, USA) and a mixture of antimycin and rotenone (1.25 and 2.5 µM, respectively; Sigma-Aldrich, St. Louis, MO, USA). During the assay, each cycle included 3 min long data acquisition points with a 3-min mixing and a 5-min waiting period in between. In all experiments, there were six measurements at the basal level and three measurements after injections of each of the metabolic inhibitors. At the end of the assay, ECAR and OCR measurements were normalised to cell numbers.

### 4.4. Biochemical Assays for Extracellular Transport Rates Measurements

We measured changes in glucose and lactate media concentration taking 100 µL of spent culture medium at the respective time points using a blood gas analyser (ABL90 FLEX, Radiometer MedicalApS, Brønshøj, Denmark). The concentration of pyruvate and a panel of amino acids were measured with their respective enzymatic assay kits (Megazyme, Wicklow, Ireland) following the microplate assay procedures supplied by the manufacturer. Absorbance was measured using a microplate reader (Spectramax M3, Molecular Devices, Sunnyvale, CA, USA).

### 4.5. Electron Microscope Imaging

Cells were grown on glass coverslips (12 mm size, Heinz Herenz) in 24 well plates and following catecholamine treatment for 4 h they were fixed with 300 µL 2% glutaraldehyde (Ted Pella, Inc., Redding, CA, USA) in 0.1 M cacodylate buffer (pH 7.4, J.B.EM Services Inc., Dorval, QC, Canada) for 20 min. The fixative was removed, and 1 mL 0.1 M cacodylate buffer was pipetted onto the cells and the cells were refrigerated until sample preparation. Coverslips were postfixed in 1% osmium tetroxide (J.B.EM Services Inc., Dorval, QC, Canada) in 0.1 M cacodylate buffer for 1 h, followed by two washes with 0.1 M cacodylate buffer for 3 min each. 

Cells were dehydrated in an ethanol series: once with 70% ethanol for 1 min, once with 96% ethanol for 1 min and twice with absolute ethanol for 1 min. Each coverslip was dipped in acetone for a few seconds and placed on an aluminium dish (Sigma-Aldrich, St. Louis, MO, USA). Immediately after, a drop of resin (Spurr Resin, Ted Pella, Inc., Redding, CA, USA) was poured on top of the coverslip covering the cells. 

A gelatine capsule was filled with resin and put upside down on top of the cells to create a block. Samples were incubated at room temperature for 2 h to allow the resin to infiltrate the cells. The blocks were then baked overnight in a 70 °C oven. After cooling down, resin blocks were separated from coverslips by dipping them for a few seconds in liquid nitrogen. Blocks were trimmed with a trimming tool (Leica EM Trim2), and then ultra-thin (100 nm) sections were cut with a diamond knife (45° Diatome) on an Ultramicrotome (Leica EM UC7) and placed on copper grids (Ted Pella, Inc.). Grids were post-stained with uranyl acetate (0.5% in distilled water, (J.B. EM Services Inc.) for 30 min and with lead citrate (3% Ultrostain II, Leica) for 1 min. Sections on grids were imaged using a JEM-1400PLUS PL Transmission Electron Microscope.

### 4.6. Metabolomics

We processed 200 µL of spent growth media by adding 30 µL of internal standard (IS) mixture comprised of 19:0 lyso phosphocholine (50 µg/mL), adenine-1,3-^15^N_2_ (50 µg/mL), L-alanine-d4 (500 µg/mL), AMP-^13^C_10_,^15^N_5_ (50 µg/mL), L-arginine-^13^C_6_ (50 µg/mL), L-carnitine-d9-^15^N_4_ (20 µg/mL), citrate-^13^C_6_ (50 µg/mL), L-cysteine-^13^C_3_,^15^N (50 µg/mL), D-glucose-^13^C_6_ (2100 µg/mL), L-glutamic acid-d5 (30 µg/mL), L-glutamine-^13^C_5_ (50 µg/mL), L-lysine-d4-^15^N_2_ (90 µg/mL), L-phenylalanine-d2 (70 µg/mL), succinic acid-d4 (50 µg/mL) and 0.5 mL of ice-cold MeOH. The samples were vortexed and centrifuged (15,000× *g*, 4 °C, 15 min). Supernatant was transferred into a new tube and then dried and stored at −80 °C until analysis. 

Samples were reconstituted in 600 µL of H_2_O:acetonitrile (50:50) and filtered by using a Pierce protein 96 well precipitation plate by centrifugation (2000× *g*, 4 °C, 15 min) prior to liquid chromatography–mass spectrometry (LC–MS) analysis. Pooled quality control samples were prepared by pooling 20 µL for each processed sample. 

Cell metabolic extracts were prepared by methanol extraction on ice. Following removal of growth media, cells were washed two times with PBS and 1 mL of 70% ice-cold methanol and 30 µL of IS mixture was added onto cells. Following cell scraping, the sample was transferred into an Eppendorf tube. The procedure was then repeated, and the extracts were combined. The extract was vortexed for 5 min and then centrifuged at 30,000 rpm for 30 min at 4 °C. The supernatant was then dried down and reconstituted in 300 µL H_2_O:acetonitrile (1:1) and vortexed until fully dissolved and filtered using a 96-well protein precipitation plate (Pierce) by centrifugation at 2000 rpm for 30 min. Samples were stored at −80 °C before LC–MS analysis.

LC–MS analysis was performed as previously described [66,67]. Briefly, the instrumentation used was an ACQUITY UPLC system (UPLC ACQUITY, Waters Corporation, Milford, MA, USA) coupled to a qTOF mass spectrometer (Synapt G2 HDMS, Waters Corporation, Manchester, UK) with an electrospray interface. The gradient chromatographic separation was performed on an ACQUITY BEH Amide (2.1 mm × 150 mm, 1.7 µm particle size, Waters Corporation) at 45 °C. 

Mobile phase A was acetonitrile + 0.1% formic acid and mobile phase B was H_2_O + 0.1% formic acid. The injection volume was 7.5 µL, the flow rate was 0.4 mL/min. The following gradient pattern was used: 0 min 99% A; 0.1 min 99% A; 6 min 40% A; 8 min 60% A; 8.5 min 99% A; and 14 min 99% A. Data were acquired in negative mode. The capillary and cone voltage were 1.5 kV and 30 V, respectively. The source and desolvation temperatures were 120 and 500 °C, respectively, and the desolvation gas flow was 800 L/h. 

MassLynx raw files were converted to mzData files using ProteoWizard Toolkit [68]. XCMS [69] was used for automatic peak-picking (centWave) and retention time alignment (OBI-Warp). Normalization of feature intensities to the added IS mixture was achieved using the R-package NormalizeMets [70]. 

Known metabolites were annotated from in-house standard libraries [66]. Unknown *m*/*z* features were called where possible from their LC–MS/MS fragmentation spectra following comparison to the METLIN database [71].

In order to measure and estimate ^13^C and ^15^N label incorporation, integration of targeted compound peaks identified from in-house standard libraries [66] and their isotopologues peaks was performed using TargetLynx (Waters). Raw data were corrected for the natural abundance of ^13^C or ^15^N isotopes using IsoCor [72] resulting in the isotopes that exceed natural abundance and the corrected isotopologue distribution.

### 4.7. Statistical Analysis

In general, we presented our data as the mean ± standard deviation when appropriate. We tested sample differences using Mann–Whitney *U* tests. We considered statistical significance under *p* < 0.05. We computed the Pearson correlation to determine associations between the studied variables.

With respect to dose-responding metabolites analysis, we first used quantile normalization to address batch variability. We determined treatment dose-responding features as those that had a significant (Wald test; *p* < 0.05) association (using function scipy.stats.linregress) with treatment and an overall treatment fold-change (FC) response greater than two (abs log_2_ FC > 1). We identified metabolite clusters using hierarchical clustering (using the seaborn.clustermap function and complete and cosine as options for the linkage method and distance metric, respectively). We identified the optimal clustering partition using the Davies–Bouldin index. Code to reproduce all computational analysis is available in the GitHub repository https://github.com/adelomana/HUVECs, accessed on 11 March 2022.

## Figures and Tables

**Figure 1 ijms-23-03162-f001:**
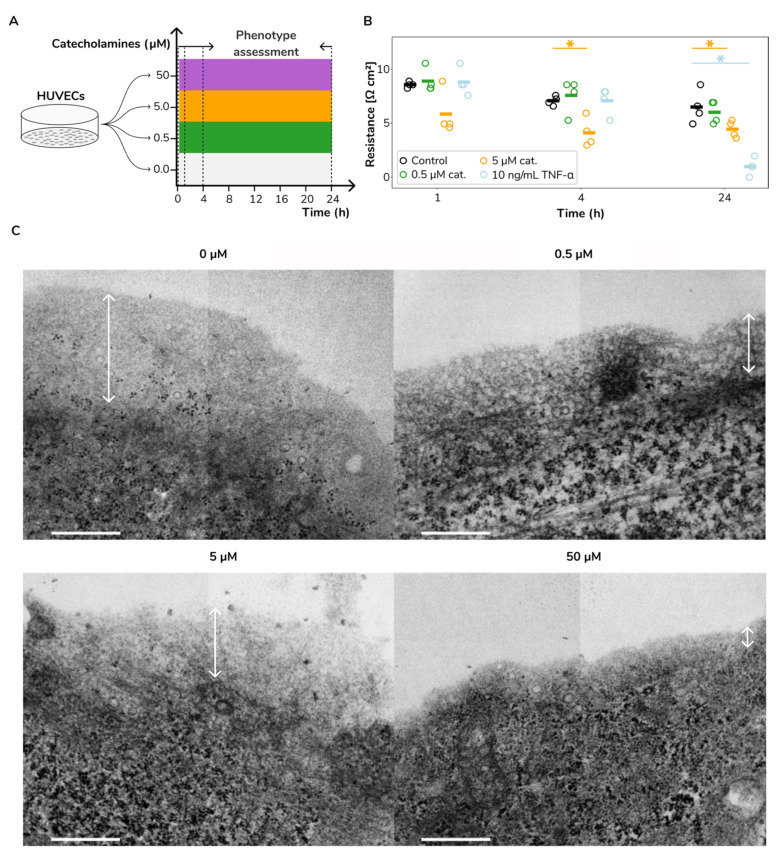
**Catecholamine stimulation leads to increased vascular permeability and glycocalyx loss**. (**A**) Overall experimental design. (**B**). Cell culture permeability measurements at 1, 4 and 24 h after treatment. Stars represent significant differences (Mann–Whitney *U* test; *p* < 0.05) between treated cultures (green, 0.5 µM equimolar adrenaline and noradrenaline; orange, 5 µM equimolar adrenaline and noradrenaline; and sky blue, 10 ng/mL TNF-α) and their corresponding time references (denoted in black colour). (**C**). TEM images of the eGC following 4 h of adrenaline and noradrenaline equimolar catecholamine stimulation (0, 0.5, 5 and 50 µM). We observed a substantial glycocalyx loss upon catecholamine stimulation. The reference bar indicates 500 nm in length and magnification of 20,000×. Double-headed white arrows highlight glycocalyx thickness.

**Figure 2 ijms-23-03162-f002:**
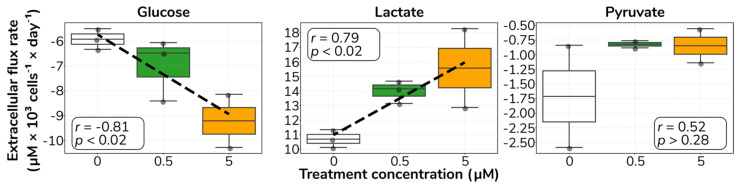
**Catecholamine stimulation increases glucose uptake and lactate secretion**. Distribution of measured extracellular transport rates for glucose, lactate and pyruvate. Positive rates indicate secretion; negative rates indicate uptake. Dashed lines denote significant correlations. White inner boxes display Pearson correlation coefficient (*r*) and *p* value (*p*).

**Figure 3 ijms-23-03162-f003:**
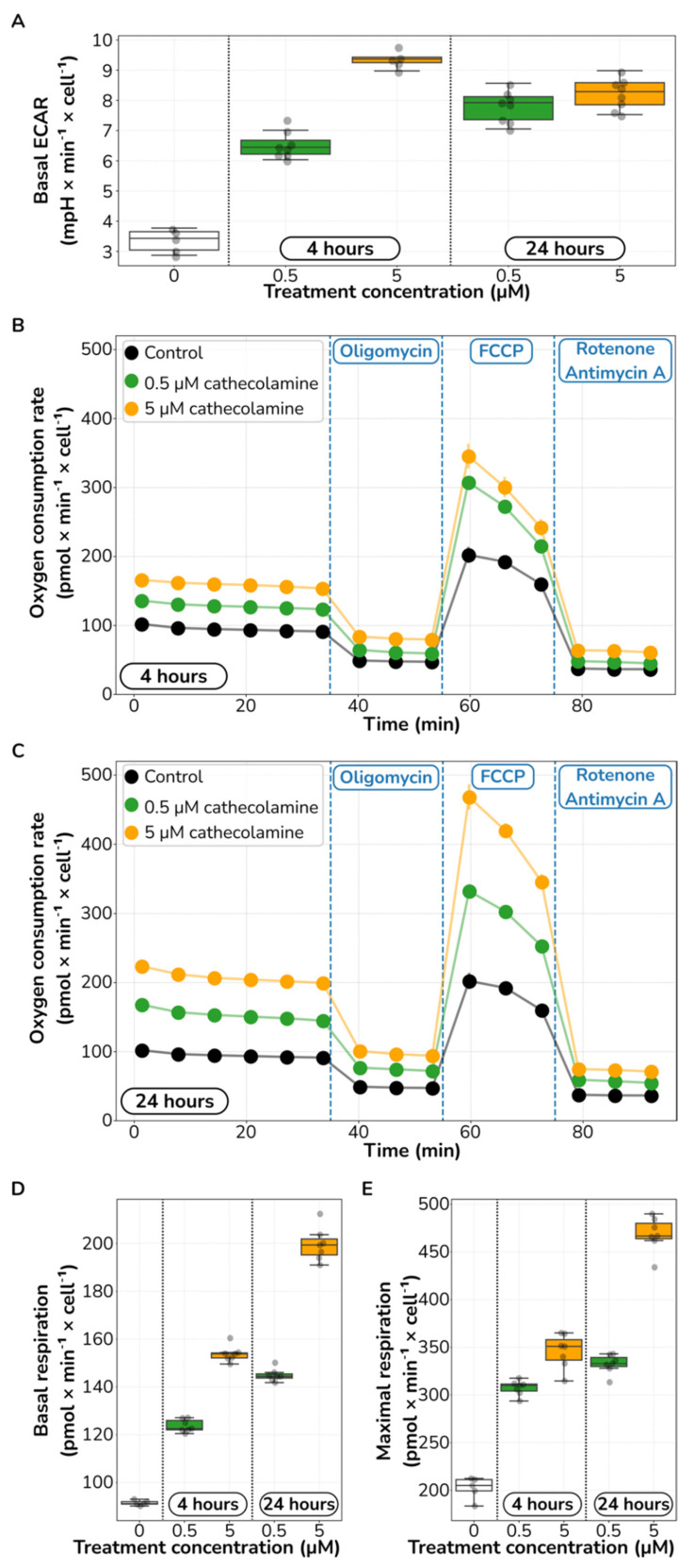
**Catecholamine stimulation increases glucose consumption and aerobic respiration**. (**A**) Basal levels of extracellular acidification rate (ECAR) in HUVECs stimulated for 4 and 24 h with different catecholamine concentrations. (**B**,**C**) Oxygen consumption rate (OCR) time profiles in HUVECs stimulated for 4 h (**B**) and 24 h (**C**) with different catecholamine concentrations. FCCP, carbonyl cyanide-*p*-trifluoromethoxyphenylhydrazone. (**D**,**E**) Effect of treatment on basal (**D**) and maximal (**E**) OCR in HUVECs.

**Figure 4 ijms-23-03162-f004:**
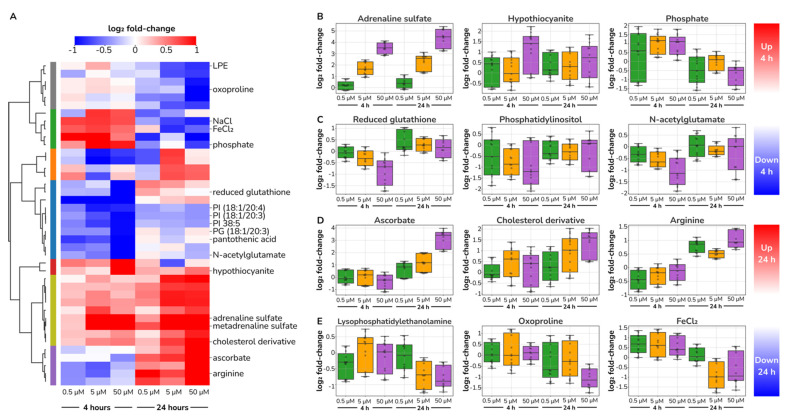
**Catecholamine stimulation alters the glutathione-ascorbate cycle and the nitric oxide biosynthesis pathway**. (**A**) Heatmap of the relative change (log_2_ fold change with respect to unstimulated cells) of the set of 46 dose–response metabolic features. Across time and catecholamine concentration (columns), metabolic features (rows) form seven clusters highlighted by a colour bar and dendrogram tree on the left. PI, phosphatidylinositol; PG, phosphatidylglycerol. (**B**–**E**) Measured metabolite fold-change intensity with respect to unstimulated cells at 4 and 24 h for different catecholamine concentrations.

**Figure 5 ijms-23-03162-f005:**
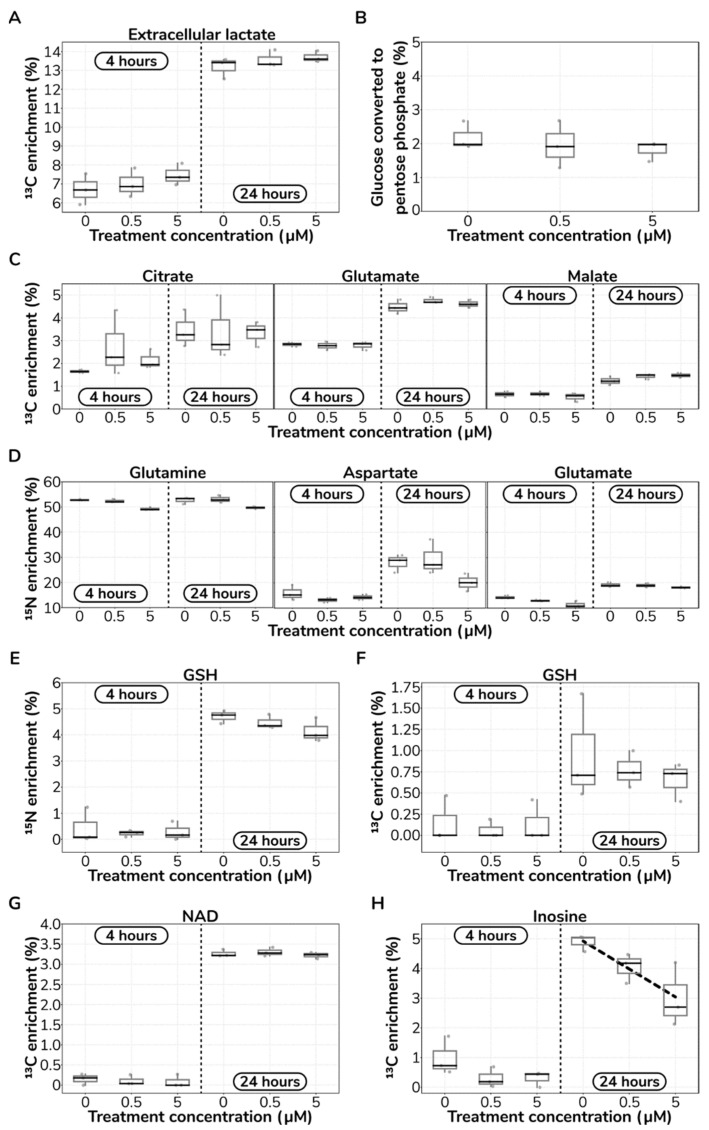
**Catecholamine stimulation does not rewire central carbon and nitrogen metabolism**. (**A**) Measured mean isotopic ^13^C enrichment in extracellular lactate derived from glucose-1,2-^13^C_2_. (**B**) Measured ratio of glucose converted into pentose phosphate. (**C**) Measured mean isotopic ^13^C enrichment in TCA cycle metabolites derived from glucose-1,2-^13^C_2_. (**D**) Measured mean isotopic ^15^N enrichment in selected metabolites derived from glutamine-^15^N_2_. (**E**) Measured mean isotopic ^15^N enrichment in glutathione derived from glutamine-^15^N_2_. (**F**) Measured mean isotopic ^13^C enrichment in glutathione derived from glucose-1,2-^13^C_2_. (**G**) Measured mean isotopic ^13^C enrichment in selected metabolites derived from glucose-1,2-^13^C_2_. (**H**) Measured mean isotopic ^13^C enrichment in selected metabolites derived from glucose-1,2-^13^C_2_. The dashed line indicates the negative relationship between mean isotopic enrichment in inosine and catecholamine concentration (*r* = −0.801; *p* < 0.01).

## Data Availability

The data that support the findings of this study are available from the corresponding author, ÓR, upon reasonable request.

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
