# Peer review of "Metabolic Response in Endothelial Cells to Catecholamine Stimulation Associated with Increased Vascular Permeability"

_ijms, 2022, doi:10.3390/ijms23063162_

Round 1
Reviewer 1 Report
The manuscript titled "Metabolic response in endothelial cells to catecholamine ...." is quite a good work that deserves to be published in IJMS after some minor corrections.
Apart from some minor errors from the technical rather than the substantive side, I do not see any major problems with it.
Below is a list of proposed or rather necessary amendments.
1. The abstract is totally changeable, firstly, shortening it and quite a lot of it, and secondly, its first part is quite confusing. Please correct it.
2. Keywords - is the last one needed?
3. Please see what this line looks like "... of what metabolic shifts occur in endothelial dysfunction. 2. Results" - was it really a pity for the authors to spend a long time preparing the manuscript. The work itself reads well, but such carelessness simply distracts you.
4. Page 3, line 97 - 98 - what is this for?
5. Page 3, lines 106 - 120 - part in one type and part in another - pity? This is also the case later in the text, and more than once.
6. Page 4 - why before figure 1 there is a caption at the top and bottom - you don't do that! And in the caption to the figers it is probably 3 different fonts. Please correct it - absolutely.
7. Figure 3 - same as above - in addition figures 3 is terribly illegible!
8. Figure 4 - same as fig 1 and fig 3 - but its legibility is even worse. You may need to consider moving parts of your drawing to reference material. Because it cannot be analyzed.
9. Discussion - only 1.5 pages - and from such a portion of results one can probably squeeze more, and much more.
10. There is totally no work summary section. Please complete this.
11. Besides, the work itself is mixed with different font styles.
To sum up, the work is quite interesting, but the style and layout itself is a misunderstanding or a lack of respect for the magazine. I do not know.
I believe that the work before the further publication process needs to be properly improved. There are no substantive errors, but its aesthetics alone leaves much to be desired. It is impossible to even evaluate some of its parts because the figers are illegible, at least some of them. After the corrections, I would like to see the works.
Reviewer 2 Report
The authors demonstrated metabolites that regulate a dose response relationship to adrenaline-noradrenaline equimolar treatment by using catecholamine-induced endothelial dysfunction model. In fact, special metabolites such as arginine and glutathione and high-stimulation of catecholamine in endothelial cells are major key regulators of homeostatic metabolic signaling for identifying traumatic injured patient. This is very important study for defining metabolic shifts occur in endothelial dysfunction. The author should present several critical points for clear data analysis and interpretation of endothelial dysfunction.
- The simulated conditions for acute critical illness (ACI) used by the authors are ambiguous. In order to suggest a specific endothelial dysfunction, research should be conducted at the level with a certain clinical outcome. Therefore, it is necessary to analyze the data in vitro and in vivo, i.e., cell and animal study.
- Methods for statistical analysis was not written in the manuscript.
- In Fig. 1., the author obtained TEM imaging to confirm the glycocalyx pattern in HUVECs, but it is recommended to add data by referring to related papers (Circ Res. 2009 Jun 5;104(11):1313-7.) in order to confirm the data and derive a definite result.
- I can't understand the interpretation of figures 2, 4, and 5 because the supplementary figure was not uploaded.
Round 2
Reviewer 1 Report
The authors improved the work quite well, although I do not agree with everything, it seems to me that the work can now be accepted.
Reviewer 2 Report
Now, it is perfect for the publication.